# The Unmet Medical Demand among China’s Urban Residents

**DOI:** 10.3390/ijerph182111708

**Published:** 2021-11-08

**Authors:** Pengfei Sheng, Tingting Yang, Tengfei Zhang

**Affiliations:** 1School of Economics, Henan University, Kaifeng 475002, China; 104753190098@henu.edu.cn; 2School of Public Finance and Taxation, Southwestern University of Finance and Economics, Chengdu 611130, China; zhangtengfei@swufe.edu.cn

**Keywords:** medical demand, the unmet medical demand, frontier demand function, Mundlak specification

## Abstract

Our work aimed to build a reasonable proxy for unmet medical demands of China’s urban residents. We combined health demand modeling and stochastic frontier analysis to produce a frontier medical demand function, which allowed us to disentangle unmet medical demands from the unobservable effects. We estimated unmet medical demands by using China’s provincial dataset that covered 2005–2018. Our estimates showed that unmet medical demand at the national level was 12.6% in 2018, and regions with high medical prices confronted more unmet medical demands than regions with moderate or low medical prices during 2005–2018. Furthermore, medical prices and education were the main factors that affected unmet medical demand; therefore, policy making should pay more attention to reducing medical costs and promoting health education.

## 1. Introduction

Human health closely affects residential living standards, and the associated human capital of health is one key element of economic growth. Residents tend to maintain their health, which drives the demand for medical services. Generally, residents deserve the same quantity and quality of medical-care services related to the same health issues, which continue unaltered by individual income levels and different locations [1]. However, the medical-care services industry suffers from asymmetric information [2], and unmet demand is common among residents [3]. The World Health Organization published the *2017 Global Monitoring Report of Tracking Universal Health Coverage* in 2017 [4]. This report declared that more than half of the world’s population could not receive basic medical services, and about 800 million residents fell into poverty because of medical spending. The unmet demand for medical-care services resulted in health inequality, resulted in income inequality, and resulted in volatile social conditions [5]. Therefore, society should change the way it mobilizes health resources at the national level. One of the key issues is identifying the unmet demand for medical-care services, which is our main goal here.

China has spent more and more to improve its public medical health care system. From the *National Health Commission of China*, China’s medical resources in 2019 reached 6.3 hospital beds per 1000 persons compared with high-income countries, which was 4.1 [6]. Meanwhile, the Chinese had a life expectancy of 77.3 years in 2019, which was an improvement compared with 71.4 years for 2000. However, China’s current medical-care system cannot satisfy a persons’ total medical demand. In *the Sixth Survey of National Health Services of China* [7], about 9% of urban hospitalized residents did not choose to be in hospitals because of financial difficulty, and 33.8% of people >65 years did not receive a conventional health check-up in 2017. Moreover, China’s fast urbanization showed that more people moved to urban areas, which created huge pressure on the medical-care system. Our study aimed to estimate unmet medical demands among China’s urban residents and provide meaningful policy implications.

Our work estimated unmet demand for medical-care services across China’s urban residents by using China’s provincial dataset for 2005–2018. Theoretically, if residents have the same demand for health, they should receive equal medical services [1]. However, residents purchased different amounts of medical-care services based on their disposable income, medical insurance, and other factors. Then, we calculated unmet medical demand by the relative distance between the actual consumption of medical-care services and the optimal demand in the economic theorem. Our findings showed that China’s unmet medical demand at the national level was 0.126 in 2018, which indicated that the current medical care system satisfied about 87.4% of total medical demand by urban residents. Meanwhile, regions with high medical prices had a higher unmet medical demand than regions with moderate and low medical prices, and all the regional unmet medical demand showed a sharp increase over 2008–2015 but a downward trend after 2015. Finally, our results also indicated that medical prices and education significantly affected unmet medical demand, and public medical care also negatively affected unmet medical demand, which might shed light on current medical policies.

Our study aimed to provide a useful approach for measuring the unmet demand for medical services. In particular, medical services defined in our work included outpatient, medical appliances, drugs, rescue transport, ward accommodation, and others for health promotion. From the individual perspective, some studies defined unmet medical demand as medical care that was unavailable when people believed it could improve his/her health necessarily [8,9,10,11,12]. However, Sanmartin et al. (2002) and Pappa et al. (2013) considered that medical resources were scarce, and the unmet medical demand was defined as the gap between the medical care expected by the residents and that was supplied by the medical system [13,14]. From the regional perspective, our study viewed that residents demanded the optimal amount of medical care in the competitive market. Moreover, the unmet medical demand was defined as the gap between medical care received in the real medical market and that demanded in the competitive medical market.

Our study contributes to the literature along three dimensions. We first discussed the health demand function of Grossman (1972) and developed a stochastic frontier function for the gross demand of residents’ medical services [15]. Second, we calculated unmet demand by the relative distance between the actual demand and the optimal demand for medical services. Moreover, third, we distinguished unmet demand from cross-sectional heterogeneity by parameterizing the mean value of the inefficient term of the stochastic frontier function, which produced a more accurate estimate of unmet medical demand. In sum, the unmet medical demand estimated in our work correlated negatively with hospitalizations per capita, which confirmed that our approach provided a useful tool to measure unmet medical demand at the regional level.

This study is organized as follows. In Section 2, we reviewed the relevant literature. Section 3 detailed the methods for estimating unmet medical demand. Section 4 defined the variables and described the dataset. In Section 5, we discussed the results. Finally, Section 6 concluded with the main findings and addressed policy implications.

## 2. Literature Review

Generally, each resident tends to maintain their health, and the medical system was supposed to follow the objective of equal use for equal needs [1]. Meanwhile, medical care was viewed as an input commodity for producing “good health” and depended on other variables in addition to health status [15]. Moreover, unmet medical demand was ubiquitous and always destroyed health equity [16].

Some studies discussed the reasons for this unmet demand. First, the economic cost had great impacts on seeking medical care, and unmet demand always occurred among residents with low income, financial constraints, or the necessity of having to spend a high share of their private funds for medical expenditures [17,18]. Second, environmental conditions influenced the depreciation rate of health capital and created an information asymmetry in health issues, which affected the unmet medical demand [19]. Third, health insurance changed residents’ willingness to pay for medical care [20], which also affected the needs and the unmet medical demand [21]. In addition, unmet medical demand was also related closely to education [22], aging [23], and social capital [18].

Several methods were developed in order model medical demand. Blundell and Windmeijer (2000) used the average waiting time to identify factors that affected medical demand and then adopted a semi-parametric selection model to specify demand [24]. Mocan et al. (2004) employed a two-part model and a discrete factor model to estimate medical demand [16]. Zhou et al. (2011) viewed the probabilities of outpatient and inpatient visits as dependent variables, then constructed a zero-truncated negative binomial equation to analyze medical demand [25]. Chaupain-Guillot and Guillot (2015) built a multilevel logistic equation to specify the factors related to unmet medical demand, which included the density of doctors, the availability of medical care, the payment for primary care physicians, and out-of-pocket payments [17]. Some other studies used a survey method to identify unmet medical demand. Mielck et al. (2009) used the question of “did you forgo any types of care because of economical cost in the last twelve months” to measure unmet medical demand [26]. Chaupain-Guillot and Guillot (2015) considered whether the respondent declared what medical care they needed, but they did not measure unmet medical demand [17]. Yoon et al. (2019) used multiple logistic regression analyses to explore the factors influencing unmet medical needs in Korea [10]. Jang et al. (2021) used logistic regression analysis to identify the effects of social networks on unmet medical needs among older adults [27].

The major contents presented in multiple studies were conveniently summarized in Table 1 from which one recognizes that the prior studies attempted to investigate the health demand for the individual. Nevertheless, unlike the previous studies, our study built a reasonable approach to measuring the unmet medical demand at the regional level, which could evaluate the allocation efficiency of the regional medical system.

Prior studies always used self-assessment of the medical care needed but which was unattained to measure unmet medical demand, and then they conducted a survey of the household to address the factors related to the unmet demand. These studies shed light on explaining the individual decision regarding the consumption of medical care, but it would be difficult to address unmet medical demands at the regional level. However, lacking the basis for the macro level, the research finding will be a biased one. Therefore, from the regional level, we should be more concerned about the regional situation as a whole and not individual circumstances. Thus, our work aimed to develop a stochastic frontier demand function for medical care and then to measure unmet medical demand.

## 3. Methodology

Within the framework of family production, Grossman (1972) provided a useful econometric model to measure health demand [15]. The model in our work also followed the above, but we rebuilt the model to address the following issues. Firstly, the health demand model in Grossman (1972) focused on residents’ decisions on health capital investment [15], but our model considered medical care as one kind of consumer goods, which could improve residents’ health. On the second, Grossman (1972) mainly answered whether an individual used health expenditure to maximize utility [15], but our model tried to estimate the amount of health care that could maintain residents’ health at the regional level. Thirdly, our study aimed to develop an econometric model to estimate unmet medical demand, which evaluated the allocation efficiency of regional medical resources and shed light on the associated public policies.

### 3.1. Model Specification

This work aimed to measure unmet medical demand, and we first began the study by specifying the optimal demand. Following the health demand model proposed by Grossman (1972) [15], medical care could also be observed as one kind of input that produces the commodity of good health. Then, optimal medical demand meant that residents purchased medical care services under the competitive medical market, and the unmet demand represented the relative distance between the actual and optimal demands. As Mocan (2004) discussed [16], medical demand was specified, as in Equation (1):(1)$$M_{it}=f(UM_{it},IM_{it},MR_{it},MP_{it},HS_{it},\varepsilon_{it})$$
where *i* and *t* are the *i* decision-making unit and the *t* period, respectively, *M_it_,* is the medical demand, *UM_it_* is the unmet demand, and *ε_it_* is the medical care resulting from unobserved group factors. The variables of income *(IM_it_*) and health condition (*HS_it_*) describe residents’ demand, the medical resources (*MR_it_*) indicate the available medical supply, and we also control the medical price (*MP_it_*) in Equation (1).

Prior studies suggested that unmet medical demand was related closely to Education (*ED_it_*), Aging (*AG_it_*), Medical insurance (*MI_it_*), Medical price (*MP_it_*), and availability of Public health care (*PM_it_*) [17,25,28,29]. Therefore, we set the unmet demand as in Equation (2):(2)$$UM_{it}=f(ED_{it},AG_{it},MI_{it},MP_{it},PM_{it},\upsilon_{it})$$
where *υ**_it_* shows the unmet demand that results from unobserved group factors.

### 3.2. Econometric Specification

Equation (1) implied that the associated variables determined the optimal medical demand, and Equation (2) formulated unmet demand. However, both equations could not be estimated directly because unmet demand was unobserved. However, it was practical to use the stochastic frontier analysis method to combine Equations (1) and (2) as follows [30,31].
(3)Mit=α+β1IMit+β2MRit+β3MPit+β4HSit+φitφit=μit+εit;εit∼iid(0,σε2); μit=UMit(Mit+UMit)=exp(−μit)

Equation (3) distinguishes the error term (*φ_it_*) between the random disturbance term (*ε_it_*) and the non-random disturbance term (*μ_it_*). The term *ε_it_* was used to capture the effects of the unobservable factors, and *μ_it_* was assumed to follow the half-normal distribution to denote unmet medical demand [30].

The main problem in estimating Equation (3) was how to specify term *μ_it_*. The first approach assumed that *μ_it_* was time-invariant, and the fixed-effects method was used to estimate Equation (3) [32]. However, the time-invariant assumption was always questioned and was not suitable for long-panel datasets. The second approach allowed *μ_it_* to be individual-specified, then used the modified fixed and random effects to derive the estimate [33], but this method was always impracticable because of overly estimated parameters. Finally, the third approach set *μ_it_* as time-varying and then aimed to disentangle the time-varying *μ_it_* from the individual-specific and time-invariant heterogeneous effects [34]. However, this method also produced biased estimates because of the so-called incidental parameters.

Furthermore, theoretically, the unobservable effects correlated strongly with the explanatory variables [35]; thus, our work used the Mundlak term to specify random term *ε_it_*. [36] In detail, the Mundlak specification parameterized the random term with the period means of the explanatory variables of Equation (3). The Mundlak specification was modeled in an auxiliary equation as follows:(4)$$\varepsilon_{it}=BX_{i}\theta +\delta_{it};BX_{i}=(\sum_{t=1}^{T}X_{it})/T;\delta_{it}∼N(0,\sigma^{2})$$
where *BX_i_* is the vector of the period means of the explanatory variables, *θ* denotes the associated coefficients vector, and *δ_it_* is the white noise term. Herein, *BX_i_θ* was built to capture the non-random effects that did not belong to the inefficient term *μ_it_*. Furthermore, because some explanatory variables in Equation (3) also affected unmet medical demand, Equation (4) excluded these variables.

It was practical to estimate Equation (3) and then to parameterize the unmet demand (the two-step approach). However, the variables that affected the inefficient term shifted and scaled the frontier function and the distribution of the inefficient term; thus, the two-step approach always resulted in biased estimates [37,38]. Thus, we parameterized the mean of unmet medical demand and built the following simultaneous equations.
(5)$$\begin{array}{l} M_{it}=\alpha +\beta_{1}IM_{it}+\beta_{2}MR_{it}+\beta_{3}MP_{it}+\beta_{4}HS_{it}+\beta_{5}PM_{it}+\varphi_{it};\\ \varphi_{it}=\mu_{it}+\varepsilon_{it};\;\\ \mu_{it}=\alpha_{\mu }+\kappa_{1}ED_{it}+\kappa_{2}AG_{it}+\kappa_{3}MI_{it}+\kappa_{4}MP_{it}+\kappa_{5}PM_{it}+\upsilon_{it};\\ \varepsilon_{it}=BX_{i}\theta +\delta_{it};BX_{i}=(\sum_{t=1}^{T}X_{it})/T;\\ \delta_{it}∼N(0,\sigma_{\varepsilon }^{2});\upsilon_{it}∼N(0,\sigma_{\upsilon }^{2}). \end{array}$$

## 4. Data

In this work, we used a panel dataset of China’s 30 provinces for 2005–2018. On the other hand, as of different statistical calibers, the sample did not include the provinces of Tibet, Hong Kong, Macao, and Taiwan, and the dataset collected variable information for permanent urban residents. Meanwhile, some variables were measured by using monetary value, and the constant prices for 2005 were used to weigh the measurements. In detail, the data sources included *China Urban life and Price Yearbook*, *Yearbook of China Insurance*, *China Statistical Yearbook*, *China Health Statistical Yearbook*, and *China Population and Employment Yearbook* [39,40,41,42,43]. Although the above dataset was collected from different publications, the associated data was generated by China’s annual National Household Income and Living Expenditure Survey. The National Bureau of Statistics of China conducted the above survey and used a multistage sampling method to produce the samples. Meanwhile, each province of China adopted the same survey scheme; then, the dataset was relatively accurate and of great statistical meaning. However, some indicators might not be updated with the market change. For instance, some nourishments also enlarged residents’ health expenditure, but the indicator did not include this expenditure. In the end, the dataset could not analyze individual health demand but was workable for addressing health demand at the regional level.

The frontier health demand model (Equation (1)) contained variables of medical care, income, medical resources, medical price, and health condition. First, health expenditure always referred to the spending on medical appliances, drugs, and medical services; then, we measured medical demand by hospitalization times, which was calculated by the frequency of residents’ health expenditures on medical price. Second, disposable income per resident was the main index for measuring the resident’s income level, which refers to the sum of residents’ final consumption expenditure and savings [44]. Disposable income included residents’ wages and salaries, net business income, and net income from properties and fiscal transfers, and then income was measured by disposable income per urban resident. Third, because the medical bed was the core resource that satisfied a resident’s health demand, we used medical beds per capita to measure medical resources. In addition, it was reasonable to measure medical price by the medical fee per outpatient, which included the doctors’ diagnosis, usage of medical equipment, medical devices, or other forms of treatment, such as physical therapy or related therapeutic procedures. Finally, as the maternal mortality rate was one of the three comprehensive indicators of national health and social progress recognized in the World Health Organization, we calculated health conditions by the mortality number per 100,000 pregnant and birth-giving women. In Equation (2), four other variables remained. First, we used schooling year per resident to denote education. Second, the old were more vulnerable to disease risks, which resulted in more medical needs and health awareness, so we used the proportion of people of the total population who was >65 years to represent aging [45]. Third, the medical benefits fund reflected the amount of medical insurance paid by residents according to national regulation pay cost standard in a region, so we used the medical benefits fund per resident to represent medical insurance. Finally, public health care was measured by the fiscal expenditure in health per resident.

Table 2 outlined the summary statistics for all variables.

## 5. Results and Discussion

### 5.1. Econometric Results

We used ordinary least squares (OLS), random effects (RE), pooled stochastic frontier analysis (FRE), and the method developed in our work (MRE) to estimate the demand function for medical care (Equation (1)). The terms *u_it_* and Lambda were significant at the 1% level, which implied that *u_it_* was relatively large and could not be ignored (Table 3); therefore, OLS and RE methods produced biased results. Meanwhile, *u_it_* from the MRE estimation was lower than that from the FRE estimation, which confirmed that the Mundlak specification disentangled *u_it_* from the group’s unobservable effects. Furthermore, the link test also showed that the prediction squared exhibited insignificant power in the MRE estimation, which implied that Equation (1) had no obvious omitted variables. Therefore, the MRE method was reasonable and preferable.

First, using the MRE approach, variables that affected unmet medical demand were introduced into Equation (5) one by one, but the variables’ coefficients showed no obvious changes in sign or significance levels, which confirmed that estimates were quite robust (Table 4). Meanwhile, all variables were statistically significant in MRE estimation (4), and the link test suggested that the specification was acceptable; Lambda statistics showed that unmet medical demand was relatively large. Consequently, our work focused on the MRE (4) estimates of Table 4.

The coefficient of income on medical demand was positive and significant, although estimates indicated that medical price affected that negatively. In detail, both the absolute values of the two coefficients were <1, which implied that medical care was a basic necessity of life [46]. Meanwhile, health conditions produced a positive effect on medical demand because residents with poor health always needed more medical care services. The coefficient of medical resources was 0.256 and significant at the 1% level, which showed that a 1% increase in medical resources raised medical demand by 0.256%. In addition, some variables in the auxiliary equation were statistically significant, which suggested that the unobservable effects correlated with the explanatory variables closely, but the point estimates had no particular meaning [47].

From the MRE (4) estimates, we also investigated variables that affected unmet medical demand. First, the coefficient of medical price was 5.8, which was significant at the 1% level. Then, our estimates suggested that unmet medical demand was price-elastic, and a 1% increase in medical price was linked to a 5.8% increase in unmet medical demand. Furthermore, the point estimate of education was negative and significant, which indicated that educated people attached more importance to their health and decreased unmet medical demand [48]. The estimate also showed that aging and medical insurance negatively affected unmet medical demand because aging persons always paid particular attention to health issues, and medical insurance lowered medical costs. Finally, public medical care positively affected unmet medical demand, possibly because the public medical care system supplied basic medical care and lowered residents’ attention to particular health issues. In sum, the above estimates were either explained by the basic economic theory or confirmed by prior studies; therefore, it was reasonable to use MRE (4) estimations to measure unmet medical demand.

### 5.2. Unmet Medical Demand

Due to the fact that medical prices were the main factor that affected unmet medical demand, we calculated provincial average medical prices during 2005–2018. Then, we divided China’s provinces into three regions: regions with high medical prices, regions with moderate medical prices, and regions with low medical prices. The regions with high medical prices included eight provinces: Beijing, Tianjin, Liaoning, Shanghai, Jiangsu, Hunan, Guangdong, and Chongqing. In particular, this region was well-developed, which took a 39.07% share of China’s GDP and a 29.17% share of China’s urban residents in 2018. Furthermore, this region was also rich in medical resources, and it has 727 tertiary hospitals and accounted for 38.10% of China’s certified doctors. The region with moderate medical prices contained 14 provinces: Hebei, Shanxi, Inner Mongolia, Jilin, Heilongjiang, Zhejiang, Anhui, Fujian, Jiangxi, Shandong Hubei, Hainan, Sichuan, and Shannxi. Finally, the region with low medical prices included eight provinces: Henan, Guangxi, Guizhou, Yunnan, Gansu, Qinghai, Ningxia, and Xinjiang. The total GDP of the moderate-medical prices region in 2018 reached 3.018 trillion RMB compared with low-medical prices region, which comprised 1.662 trillion RMB. Meanwhile, the region with moderate medical prices had an average urban population of 29.524 million and 33,775 health professionals in 2018. The low-medical prices region had an average urban population of 18.398 million and 24,014 health professionals in 2018.

We calculated the average unmet medical demand for China and regions weighted by permanent urban residents (Figure 1). Unmet medical demand was 12.6% for China in 2018, implying that about 12.6% of residents’ medical demand was unsatisfied. Regions with high medical prices had more unmet demands than regions with moderate and low medical prices during 2005–2018. There were several reasons that explained the higher unmet medical demand in the region with high medical prices. At first, this region was well-developed, and the high income would increase residents’ health demand while the medical system could not satisfy the demanded medical care service. Secondly, China’s medical system still had a method to carry out the hierarchical medical system, and then the medical resources were always underutilized. Moreover, few residents received meaningful health education, and health demand was always driven by the rising income or the available medical resources rather than by critical disease needs.

During 2008–2013, unmet medical demand showed a sharply increasing trend at the national and regional levels. Meanwhile, China has improved the rural cooperative medical system and an urban basic medical insurance system since 2008, but the medical resources lagged behind the increasing medical demand, which resulted in increasing unmet medical demand. In particular, there were 858,015 community medical institutions and 24,260 community healthcare centers in 2005, but the two numbers increased to 943,639, and 34,997 in 2018, respectively. Therefore, unmet medical demand has declined since 2015. In sum, medical prices mainly increased residents’ unmet medical demand, but the growing medical resources lowered the demand in China.

In 2018, Tianjin, Shanghai, and Zhejiang performed best for unmet medical demand, and their unmet demands were < 5% (Figure 2). In contrast, Jiangxi, Fujian, Guangdong, Beijing, and Hainan were ranked as the top five provinces, and their unmet medical demands were >20%. During 2005–2018, six provinces decreased their unmet medical demand by >10%: Shanghai, Guizhou, Jiangsu, Hainan, Ningxia, and Hubei. However, some provinces also experienced an upward trend in unmet medical demand, including Guangdong, Fujian, Chongqing, Jiangxi, Yunnan, Beijing, and Gansu. Moreover, developed provinces always experienced high unmet medical demands, such as Fujian, Guangdong, Beijing, and Hainan. By comparison, those in developing provinces had lower unmet medical demands, such as Gansu, Henan, Shannxi, and Qinghai. Furthermore, the main reason was that developed provinces often experienced high medical prices, but developing provinces enjoyed low medical prices. In sum, we concluded from the provincial perspective that high medical prices contributed more to unmet medical demand than did economic development.

Conventional wisdom suggested that hospitalization per capita was a reasonable indicator of residents’ medical demand. Thus, if the method in our work was a useful tool for measuring unmet medical demand, then there should be negative correlations between unmet medical demand and hospitalization per capita. The overall correlation coefficient between the estimated “unmet medical demand” and hospitalization per capita was −0.776, which was an obvious negative correlation. Furthermore, from a provincial perspective, negative correlations occurred among 27 provinces, but only three had a positive correlations. Therefore, the above results confirmed that the method in this work was a reasonable proxy for unmet medical demand.

## 6. Conclusions

Our work attempted to build a reasonable approach to measure unmet medical demand using China’s provincial dataset for 2005–2018. The approach combined health demand modeling and stochastic frontier analysis, which allowed us to build a frontier demand function for medical care. Then, we calculated the relative distance from the actual medical demand to the optimal medical demand, which was defined as a proxy for unmet medical demand. In 2018, about 12.6% of residents’ medical demand was unsatisfied at the national level. Meanwhile, during 2005–2018, unmet medical demand rose and then fell at the national and regional levels. Moreover, medical prices mainly affected unmet medical demand, and the region with high medical prices always experienced higher unmet medical demand than regions with low medical prices.

Our findings provide an economic rationale for policy making. First, the approach in our study could help policy makers identify whether the allocation of medical resources was effective at the regional level. Second, unmet medical demand would be increased with medical resources, and then it was workable to adopt the proper hierarchical medical system, which could avoid the underutilization of medical resources. Finally, public policy could attach more to health education, which was able to reduce unnecessary medical demand and alleviate strains on the medical system.

Our research is subject to some limitations. First, this paper investigated unmet medical demands from a regional view for China’s provincial dataset over 2005–2018, but we could not simultaneously draw the findings for individuals in the empirical study. Second, we can use the Mundlak specification to isolate unmet medical demands from the cross-sectional effects but not fully, which may result in the unmet medical demands error being measured.

## Figures and Tables

**Figure 1 ijerph-18-11708-f001:**
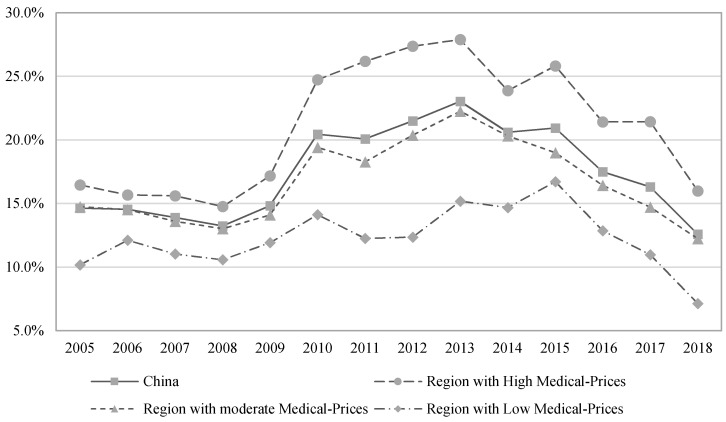
The unmet demand for China and the regions.

**Figure 2 ijerph-18-11708-f002:**
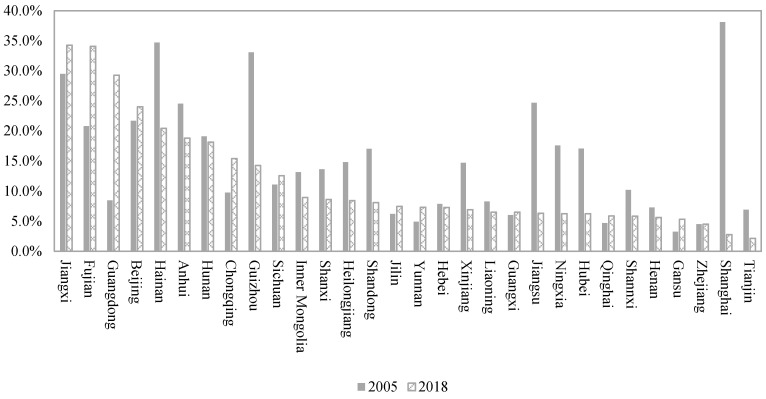
The unmet medical demand across China’s provinces.

**Table 1 ijerph-18-11708-t001:** Studies for unmet medical demand.

Study	Concept	Method	Explaining Variables
Grossman (1972) [15]	Demand for health	Conceptual Framework	Medical care, age, time inputs, goods input, oc human capital
Blundell and Wind-meijer (2000) [24]	Demand for health services at the ward level in the UK	Semi-parametric selection model	Average waiting time routine surgery in days, standardized estimated costs 1991–1992 acute care, NHS (National Health Service of the United Kingdom) hospital accessibility, general practitioner accessibility, the proportion of the 75 years and older not in nursing and residential homes, private hospital accessibility, standardized mortality ratio ages 0–74, standardized illness ratio ages 0–74, for residents in households only, the proportion of persons with head in manual class, proportion of those of pensionable age living alone, the proportion of dependents in single households, proportion of the economically active that is unemployed, and proportion of residents in households with no car
Mocan et al. (2004) [16]	Demand for medical care covered urban households	Two-part model and a discrete factor model	Price of medical care, price of food, the opportunity cost of time, age, environmental, and the variables that influence the productivity of health investment
Ariizumi (2008) [21]	Medical care demand	Conceptual Framework	Public long-term care insurance, age, health status, medical investment, level of consumption, and presence of a chronic illness
Zhou et al. (2011) [25]	Demand for healthcare across individuals from rural China	The probit regression model, zero-truncated negative binomial equation	Outpatient price, inpatient price, income, gender, age, and marriage
Pappa et al. (2013) [14]	Unmet health needs across 1000 consenting subjects	Multiple binary logistic regression analysis	Sex, age, marital status, children, education, occupation, urbanity, physician, consultations, and chronic diseases
Chaupain-Guillot and Guillot (2015) [17]	Unmet care needs across 400,000 individuals aged 16 and over	Multilevel logistic equation	Sex, age, self-perceived health, citizenship, education level, family situation, household income, housing tenure status, the existence of debts, and whether the household has a car or not
Connolly and Wren (2017) [9]	Unmet healthcare needs across 4922 households in Ireland	Multivariate logistic regression	Sex, age group, marital status, education, principal economic status, income, eligibility category, health status, and chronic illness
Yoon et al. (2019) [10]	Unmet medical needs across adults over 19 years old from Korea	Multiple logistic regression	Sex, age, education, marital status, economic activity, income, medical insurance type, private insurance, non-covered treatment, chronic disease, disability, regular exercise, pain, self-rated health status, and depression
Fiorillo (2020) [18]	Unmet needs for health care across 260,000 respondents from 14 Member States of the EU	Expanded probit model	Social capital, social support, and individual characteristics (gender, marital status, age, household size, country of birth, education, economic features, health status, and size of municipality)
Njagi et al. (2020) [11]	Unmet need for healthcare services across 33,675 households n from Kenya	Multilevel regression model	Gender, age, education level, employment status, type of service, self-rated health, chronic illness, insurance status, household size, residence, and wealth index
Jang et al. (2021) [27]	Unmet medical needs across 2281 older adults with limited IADL from Koreans	Logistic regression analysis	Age, gender, educational level, household income, number of chronic diseases, living arrangement, contact with friend and neighbor, social activity, emotional support, instrumental support, physical support, and financial support
Jung and Ha (2021) [12]	Unmet healthcare needs across 26,598 participants aged 19 years and older from Korea	Multiple logistic regression models	Age, marital status, family member, education level, region, employment, income, occupation, medical insurance type, private insurance, smoking history, alcohol consumption, body mass index, exercise, self-rated health status, stress level, pain, and depression

**Table 2 ijerph-18-11708-t002:** Summary statistics of variables.

	Mean	Maximum	Minimum	Coefficient of Variance
Medical Demand	1.775	2.464	1.003	0.144
Income	9.730	10.826	8.986	0.040
Medical Price	4.966	5.999	3.993	0.060
Health Condition	2.790	4.760	0.095	0.263
Medical Resources	1.819	3.311	0.920	0.225
Education	2.166	2.530	1.853	0.052
Aging	2.190	2.728	1.449	0.106
Health Insurance	7.434	8.775	6.263	0.061
Public Health Care	5.875	7.428	3.655	0.138

**Table 3 ijerph-18-11708-t003:** Test for the MRE estimations.

	OLS	RE	FRE		MRE	
			Main Equation	Auxiliary Equation	Main Equation	Auxiliary Equation
Income	0.359 ***	0.785 ***	0.341 ***		0.306 ***	−3.040
	(0.065)	(0.063)	(0.060)		(0.059)	(2.154)
Medical Price	−0.519 ***	−0.601 ***	−0.495 ***		−0.447 ***	
	(0.060)	(0.085)	(0.055)		(0.065)	
Health Condition	0.018	0.064 ***	0.022		0.008	0.058
	(0.023)	(0.020)	(0.022)		(0.022)	(0.761)
Medical Resources	0.276 ***	0.008	0.300 ***		0.292 ***	−0.959
	(0.033)	(0.037)	(0.033)		(0.032)	(0.905)
Constant	0.308	−3.069 ***	0.534	−4.310 ***	0.685	26.726
	(0.536)	(0.441)	(0.498)	(0.278)	(0.479)	(22.479)
*u_it_*			0.285 ***	0.276 ***		
			(0.025)	(0.021)		
Lambda			2.494 ***	2.490 ***		
			(0.039)	(0.040)		
Prediction Squared	−0.071		−0.250	−0.271		
	(0.344)		(0.320)	(0.355)		

Notes: the standard errors are in parentheses; *** *p* < 0.01, ** *p* < 0.05, * *p* < 0.1.

**Table 4 ijerph-18-11708-t004:** Estimations for medical demand.

	MRE (1)		MRE (2)		MRE (3)		MRE (4)	
	Main Equation	Auxiliary Equation	Main Equation	Auxiliary Equation	Main Equation	Auxiliary Equation	Main Equation	Auxiliary Equation
Medical Demand								
Income	0.297 ***	−5.358 ***	0.422 ***	−1.727 **	0.339 ***	−1.529 *	0.247 ***	−4.400 **
	(0.054)	(1.939)	(0.063)	(0.762)	(0.073)	(0.781)	(0.055)	(2.003)
Medical Price	−0.264 ***		−0.494 ***		−0.393 ***		−0.178 **	
	(0.074)		(0.077)		(0.084)		(0.073)	
Health Conditions	0.035 *	−0.389	0.056 **	−0.772 **	0.049 **	−0.686 **	0.030 *	0.071
	(0.021)	(0.634)	(0.022)	(0.332)	(0.022)	(0.327)	(0.020)	(0.627)
Medical Resources	0.267 ***	−0.989	0.250 ***	0.902 *	0.235 ***	0.959 *	0.256 ***	−0.174
	(0.030)	(0.791)	(0.032)	(0.495)	(0.032)	(0.498)	(0.034)	(0.683)
Constant	−0.190	50.491 **	−0.402	13.809 *	−0.047	11.517	−0.098	38.413 *
	(0.510)	(19.988)	(0.565)	(7.862)	(0.574)	(8.122)	(0.484)	(20.720)
The Unmet Medical Demand								
Medical Price	3.498 ***		6.718 **		8.550 ***		5.800 ***	
	(0.728)		(2.835)		(2.646)		(1.029)	
Education	−6.047 ***		−15.967 ***		−14.840 ***		−6.797 ***	
	(1.366)		(4.958)		(3.989)		(1.486)	
Aging			−4.181 ***		−3.888 ***		−1.303 ***	
			(1.359)		(1.231)		(0.401)	
Medical Insurance					−1.847 **		−2.095 ***	
					(0.817)		(0.490)	
Public Medical Care							0.496 *	
							(0.274)	
Constant	−7.073 ***		5.565		7.163		−1.600	
	(2.251)		(7.158)		(5.586)		(2.607)	
Lambda	2.285 ***		2.408 ***		2.500 ***		2.114 ***	
	(0.040)		(0.040)		(0.039)		(0.041)	
Prediction Squared	−0.212		−0.009		−0.156		−0.269	
	(0.427)		(0.345)		(0.446)		(0.503)	

Notes: the standard errors were in parentheses; *** *p* < 0.01, ** *p* < 0.05, * *p* < 0.1.

## Data Availability

The data was collected from World Bank and the National Bureau of Statistics of China.

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
