# Peer review of "The Unmet Medical Demand among China’s Urban Residents"

_ijerph, 2021, doi:10.3390/ijerph182111708_

Round 1

Reviewer 1 Report

Dear authors, the work, although well written, has more of a practical rather than a scientific approach.

I think that just measuring the unmet medical demand is not a sufficient reason to support a scientific proposal.

Besides that, most of the references used in the construction of the proposal are from very old works and the approach seems to me rather lacking experience in terms of contribution.

Author Response

Dear Respected Reviewers,

Many thanks for your kind comments concerning our manuscript entitled "The Unmet Medical Demand among China's Urban Residents" (ID: ijerph-1415182). Those comments are valuable and helpful for improving our work and the important guideline for us. We have studied comments carefully and made a major revision, hoping to meet with approval. All revised portions of the manuscript were marked up using the "track changes" function. Main corrections in the paper and responses are described below.

Best Wishes

Sincerely

Comments and Revision Description (1)

  1. The work, although well written, has more of a practical rather than a scientific approach. I think that just measuring the unmet medical demand is not a sufficient reason to support a scientific proposal.

Responses: Our work attempted to build a reasonable approach measuring the unmet medical demand; then provided the economic rationale for evaluating the allocation efficiency of the regional medical system. We then highlight the main finding and the potential contribution in the introduction part. Please find the following or the introduction part.

"Our work estimated unmet demand for medical-care services across China's urban residents by using China's provincial dataset for 2005-2018. Theoretically, if residents have the same demand for health, they should get equal medical services (Morris et al., 2005). However, residents purchased different amounts of medical-care services based on their disposable income, medical insurance, and other factors. Then, we calculated unmet medical demand by the relative distance between the actual consumption of medical-care services and the optimal demand in the economic theorem. Our findings showed that China's unmet medical demand at the national level was 0.126 in 2018, which indicated that the current medical-care system satisfied about 87.4% of total medical demand by urban residents. Meanwhile, regions with high medical prices had a higher unmet medi-cal demand than regions with moderate and low medical prices, and all the regional un-met medical demand showed a sharp increase over 20182008-2015, but a downward trend after 2015. Finally, our results also indicated that medical prices and education sig-nificantly affected unmet medical demand, and public medical care also negatively af-fected that, which might shed light on current medical policies.

Our work aimed to provide a useful approach to measuring the unmet demand for medical services. In particular, medical services defined in our work included outpatient, medical appliances, drugs, rescue transport, ward accommodation, and others for health promotion. From the individual perspective, some studies defined unmet medical demand as the medical care which was unavailable when people believed it could improve his/her health necessary (Reeves et al., 2015; Connolly and Wren, 2017; Yoon et al., 2019; Njagi et al., 2020; Jung and Ha, 2021). However, Sanmartin et al. (2002) and Pappa et al. (2013) considered that medical resources were scarce, and the unmet medical demand was defined as the gap between the medical care expected by the residents and that sup-plied by the medical system. From the regional perspective, our work viewed that residents demanded the optimal amount of medical care in the competitive market. And then, the unmet medical demand was defined as the gap between the medical care received in the real medical market, and that demanded in the competitive medical market.

Our work contributes to the literature along three dimensions. We first discussed the health demand function of Grossman (1972) and developed a stochastic frontier function for the gross demand of residents' medical services. Second, we calculated unmet demand by the relative distance between the actual demand and the optimal demand for medical services. Moreover, third, we distinguished unmet demand from the cross-sectional heterogeneity by parameterizing the mean value of the inefficient term of the stochastic frontier function, which produced a more accurate estimate of unmet medical demand. In sum, the unmet medical demand estimated in our work correlated negatively with hospitalizations per capita, which confirmed that our approach provided a useful tool to measure unmet medical demand at the regional level."

  1. Most of the references used in the construction of the proposal are from very old works and the approach seems to me rather lacking experience in terms of contribution.

Responses: Candidly, the prior studies focused on the unmet medical need for the individual, and few addressed the issue at the regional level. Meanwhile, we also tried to analyze the existing studies and update the literature. Then, we improve the literature review and the reference list. Please see the following added literature or parts of the introduction and the literature review.

Added literature:

Yoon, Y. S., Jung, B., Kim, D., & Ha, I. H. (2019). Factors underlying unmet medical needs: a cross-sectional study. International journal of environmental research and public health, 16(13), 2391.

Jang, H. Y., Ko, Y., & Han, S.Y. (2021). The Effects of Social Networks of the Older Adults with Limited Instrumental Activities of Daily Living on Unmet Medical Needs. International Journal of Environmental Research and Public Health, 18(1), 27.

Reeves, A., McKee, M., Stuckler, D. (2015). The attack on universal health coverage in Europe: Recession, austerity and unmet needs. European Journal of Public Health, 25, 364–365.

Yoon, Y. S., Jung, B., Kim, D., & Ha, I. H. (2019). Factors underlying unmet medical needs: a cross-sectional study. International journal of environmental research and public health, 16(13), 2391.

Sanmartin, C., Houle, C., Tremblay, S., & Berthelot, J. M. (2002). Changes in unmet health care needs, Health Reports, 13, 15–21.

Pappa, E., Kontodimopoulos, N., Papadopoulos, A., Tountas, Y., & Niakas, D. (2013). Investigating unmet health needs in primary health care services in a representative sample of the Greek population, International Journal of Environmental Research and Public Health, 10, 2017–2027.

Yang, S., Wang, D., Li, W., Wang, C., Yang, X., & Lo, K. (2021). Decoupling of Elderly Healthcare Demand and Expenditure in China. Healthcare, 9(10), 1346.

Zhang, K., Kumar, G., & Skedgel, C. (2021). Towards a New Understanding of Unmet Medical Need. Applied Health Economics and Health Policy, 1-4.

Smith, S., & Connolly, S. (2020). Re-thinking unmet need for health care: introducing a dynamic perspective. Health Economics, Policy and Law, 15(4), 440–457.

Jung, B., & Ha, I. H. (2021). Determining the reasons for unmet healthcare needs in South Korea: a secondary data analysis. Health and quality of life outcomes, 19(1), 99.

Connolly, S., & Wren, M. A. (2017). Unmet healthcare needs in ireland: analysis using the eu-silc survey. Health Policy, 121(4), 434.

Njagi, P., Arsenijevic, J., & Groot, W. (2020). Cost–related unmet need for healthcare services in kenya. BMC Health Services Research, 20(1). https://doi.org/10.1186/s12913-020-05189-3

Mehrara, M., Musai, M., & Amiri, H. (2010). The relationship between health expenditure and GDP in OECD countries using PSTR. European Journal of Economics, Finance and Administrative Sciences, 24, 1450-2275.

Reviewer 2 Report

  • Use of a large dataset with multiple years is a strength of the study.
  • A clear working definition of unmet demand would be helpful in the introduction.  Is it for basic medical services that can’t be fulfilled or does it go beyond that? Section 2 goes into the various ways it has been examined in the literature, but how would authors define it for this particular purpose based on what they have reviewed.
  • Differences in terms of strengths/weaknesses in the measurement of demand would be helpful to include in section 2 to show how the current version may be addressing some of these issues and what it may not be able to answer fully using Grossman’s model.
  • Section 4- how closely does the available data reflect the “ideal” set of variables? Are there limits in what can be assessed using these datasets? Some discussion of the choice of measures to include would be helpful.
  • Authors highlight differences by province.  I’m not familiar with these regions and was wondering if authors could discuss/describe some differences by region beyond data collected in some way.  Would this be what other studies may suggest as areas with better performance overall in health care?  For example, in the United States, the “Southern” states tend to have “worse” performance on many health care quality measures for a variety of reasons.  Are the findings of provinces generally expected from prior research on them?
  • A section describing the study's practical implications would be helpful as well as strengths/limitations and areas for future research.  

Author Response

Dear Respected Reviewers,

Many thanks for your kind comments concerning our manuscript entitled "The Unmet Medical Demand among China's Urban Residents" (ID: ijerph-1415182). Those comments are valuable and helpful for improving our work and the important guideline for us. We have studied comments carefully and made a major revision, hoping to meet with approval. All revised portions of the manuscript were marked up using the "track changes" function. Main corrections in the paper and responses are described below.

Best Wishes

Sincerely

Comments and Revision Description (1)

  1. A clear working definition of unmet demand would be helpful in the introduction. Is it for basic medical services that can't be fulfilled or does it go beyond that? Section 2 goes into the various ways it has been examined in the literature, but how would authors define it for this particular purpose based on what they have reviewed.

Responses: We clarified the meaning of the unmet medical demand with the prior studies and then defined that at the regional level. Please find the following or in the third paragraph of the Introduction part.

"Our work aimed to provide a useful approach to measuring the unmet demand for medical services. In particular, medical services defined in our work included outpatient, medical appliances, drugs, rescue transport, ward accommodation, and others for health promotion. From the individual perspective, some studies defined unmet medical demand as the medical care which was unavailable when people believed it could improve his/her health necessary (Reeves et al., 2015; Connolly and Wren, 2017; Yoon et al., 2019; Njagi et al., 2020; Jung and Ha, 2021). However, Sanmartin et al. (2002) and Pappa et al. (2013) considered that medical resources were scarce, and the unmet medical demand was defined as the gap between the medical care expected by the residents and that sup-plied by the medical system. From the regional perspective, our work viewed that resi-dents demanded the optimal amount of medical care in the competitive market. And then, the unmet medical demand was defined as the gap between the medical care received in the real medical market, and that demanded in the competitive medical market.."

  1. Differences in terms of strengths/weaknesses in the measurement of demand would be helpful to include in section 2 to show how the current version may be addressing some of these issues and what it may not be able to answer fully using Grossman's model.

Responses: Our model aimed to evaluate the allocation efficiency of the regional medical system while Grossman (1972) attempted to investigate the health demand for the individual. We discussed the difference between our model and that of Grossman (1972). Please find the following or the first paragraph of the methodology part.

"Within the framework of family production, Grossman (1972) provided a useful econometric model to measure health demand. The model in our work also followed the above, but we rebuilt the model to address the following issues. Firstly, the health demand model in Grossman (1972) focused on residents' decisions on health capital investment, but our model considered medical care as one kind of consumer goods, which could improve residents' health. On the second, Grossman (1972) mainly answered whether an individual used health expenditure to maximize the utility, but our model tried to estimate the amount of health care that could maintain residents' health at the regional level. Thirdly, our work aimed to develop an econometric model to estimate unmet medical demand, which evaluated the allocation efficiency of regional medical resources and shed light on the associated public policies."

  1. Section 4- how closely does the available data reflect the "ideal" set of variables? Are there limits in what can be assessed using these datasets? Some discussion of the choice of measures to include would be helpful.

Responses: we gave more description of the dataset and then discussed the relationship between the variables and the statistical indicators. Please find the following or the data part.

" In this work, we used a panel dataset of China's 30 provinces for 2005-2018. Whereas, as of different statistical calibers, the sample did not include the provinces of Tibet, Hong Kong, Macao, and Taiwan, and the dataset collected the variable information for perma-nent urban residents. Meanwhile, some variables were measured using monetary value, and the constant prices for 2005 were used to weight the measurements. In detail, the data sources included China Urban life and Price Yearbook, Yearbook of China Insurance, China Sta-tistical Yearbook, China Health Statistical Yearbook, and China Population & Employment Year-book. Although the above dataset was collected from different publications, the associated data was generated by China's annual National Household Income and Living Expenditure Survey. National Bureau of Statistics of China conducted the above survey and used the mul-tistage sampling method to produce the samples. Meanwhile, each province of China adopted the same survey scheme; then, the dataset was relatively accurate and of great statistical meaning. However, some indicators might not be updated with the market change. For instance, some nourishments also enlarged residents' health expenditure, but the indicator did not include this expenditure. In the end, the dataset could not analyze individual health demand but was workable to address the health demand at the regional level.

The frontier health demand model (equation 1) contained variables of medical care, income, medical resources, medical price, and health condition. First, health expenditure always referred to the spending on medical appliances, drugs, and medical services; then, we measured medical demand by hospitalization times, which was calculated by the frequency of residents' health expenditures on medical price. Second, the disposable in-come per resident was the main index to measure the resident's income level, which refers to the sum of residents' final consumption expenditure and savings (Mehrara et al., 2010). The disposable income included residents' wages and salaries, net business income, and net income from properties and fiscal transfers, and then income was measured by the disposable income per urban resident. Third, because the medical bed was the core resource that satisfied a resident's health demand, we used medical beds per capita to measure medical resources. In addition, it was reasonable to measure medical price by the medical fee per outpatient, which included the doctors' diagnosis, usage of medical equipment, medical devices, or other forms of treatment, such as physical therapy or re-lated therapeutic procedures. Finally, as the maternal mortality rate was one of the three comprehensive indicators of national health and social progress recognized in the World Health Organization, we calculated health conditions by the mortality number per 100,000 pregnant and birth-giving women. In equation (2), four other variables remained. First, we used the schooling year per resident to denote education. Second, the old were more vulnerable to disease risks, which led to more medical needs and health awareness, so we used the proportion of people of the total population who were >65 years to repre-sent aging (Yang et al., 2021). Third, the medical benefits fund reflected the amount of medical insurance paid by residents according to national regulation pay cost standard in a region, so we used the medical benefits fund per resident to represent medical insurance. Finally, public health care was measured by the fiscal expenditure in health per resident."

  1. Authors highlight differences by province. I'm not familiar with these regions and was wondering if authors could discuss/describe some differences by region beyond data collected in some way. Would this be what other studies may suggest as areas with better performance overall in health care? For example, in the United States, the "Southern" states tend to have "worse" performance on many health care quality measures for a variety of reasons. Are the findings of provinces generally expected from prior research on them?

Responses: we revised two parts of our work. The first was to describe the social-economic status of the three regions, and the second was to reason why the region with high medical prices performed not well in the unmet medical demand. Please find the following or the 5.2 part.

" Because the medical price was the main factor that affected unmet medical demand, we calculated provincial average medical prices during 2005-2018. Then, we divided China's provinces into three regions: regions with high medical prices, regions with moderate medical prices, and regions with low medical prices. The regions with high medical prices included eight provinces: Beijing, Tianjin, Liaoning, Shanghai, Jiangsu, Hunan, Guangdong, and Chongqing. In particular, this region was well-developed, which took a 39.07% share of China's GDP and a 29.17% share of China's urban residents in 2018. Furthermore, this region was also rich in medical resources, and it has 727 tertiary hospitals and accounted for 38.10% of China's certified doctors. The region with moderate medical prices contained 14 provinces: Hebei, Shanxi, Inner Mongolia, Jilin, Heilongjiang, Zhejiang, Anhui, Fujian, Jiangxi, Shandong Hubei, Hainan, Sichuan, and Shannxi. Finally, the region with low medical prices included eight provinces: Henan, Guangxi, Guizhou, Yunnan, Gansu, Qinghai, Ningxia, and Xinjiang. The total GDP of the moderate-medical prices region in 2018 reached 3.018 trillion RMB compared with the low-medical prices region, which was 1.662 trillion RMB. Meanwhile, the region with moderate medical prices had an average urban population of 29.524 million and 33775 health professionals in 2018. The low-medical prices region had an average urban population of 18.398 million and 24014 health professionals in 2018.

We calculated the average unmet medical demand for China and regions weighted by permanent urban residents (Fig. 1). Unmet medical demand was 12.6% for China in 2018, implying that about 12.6% of residents' medical demand was unsatisfied. Regions with high medical prices had more unmet demand than regions with moderate and low medical prices during 2005-2018. There were several reasons that explained the higher unmet medical demand in the region with high medical prices. At first, this region was well-developed, and the high income would increase residents' health demand while the medical system could not satisfy the demanded medical care service. Secondly, China's medical system still had a way to carry out the hierarchical medical system, and then the medical resources were always underutilized. Moreover, few residents received meaningful health education, and health demand was always driven by the rising income or the available medical resources rather than the critical disease needs."

  1. A section describing the study's practical implications would be helpful as well as strengths/limitations and areas for future research.

Responses: Many thanks for your patience. We discussed the practical implications and the limitations in the concluding parts. You can also see the following.

" Our findings provide an economic rationale for policy-making. First, the approach in our work could help policy-makers identify whether the allocation of medical resources was effective at the regional level. Second, the unmet medical demand would be increased with the medical resources, and then it was workable to adopt the proper hierarchical medical system, which could avoid the underutilization of medical resources. Finally, public policy could attach more to health education, which was able to reduce the unnec-essary medical demand and alleviate the strains on the medical system.

Our research is subject to some limitations. First, this paper investigated unmet med-ical demands from a regional view for China's provincial dataset over 2005-2018, but we could not draw the findings for individuals simultaneously in the empirical study. Sec-ond, we can use the Mundlak specification to isolate the unmet medical demands from the cross-sectional effects but not fully, which may result in the unmet medical demands error being measured."

Round 2

Reviewer 1 Report

The references used in the revised version of the work support the assertions and proposals of the work, clearly revealing the state of the art regarding the topic addressed.

Regarding the proposal itself, the authors made the contributions to Grossman's proposal more evident, presenting modifications in relation to the model of the original work.

Despite this, I think this section could be enriched with more information about the gaps in the research compared to the contributions proposed in the article.

Furthermore, it would be very interesting for the reader to be able to view this information in a compiled format. Using a table, for example, in which the approaches and gaps of each proposal would be demonstrated, allowing to easily visualize the gap in the research area of ​​the article.

Author Response

Dear Respected Reviewer,

Many thanks for your kind comments concerning our manuscript entitled "The Unmet Medical Demand among China's Urban Residents" (ID: ijerph-1415182). Those comments help improve the quality of our work. We have studied comments carefully and made the revisions, hoping to meet with approval. The revised portions of the manuscript were marked up using the "track changes" function.

Best Wishes

Sincerely

Comments and Revision Description

  1. The references used in the revised version of the work support the assertions and proposals of the

work, clearly revealing the state of the art regarding the topic addressed. Regarding the proposal itself, the authors made the contributions to Grossman's proposal more evident, presenting modifications in relation to the model of the original work. Despite this, I think this section could be enriched with more information about the gaps in the research compared to the contributions proposed in the article. Furthermore, it would be very interesting for the reader to be able to view this information in a compiled format. Using a table, for example, in which the approaches and gaps of each proposal would be demonstrated, allowing to easily visualize the gap in the research area of the article.

Responses: We sincerely accepted the suggestion and made the revision. Please find the following or the Literature Review part.

" The major contents presented in multiple studies were conveniently summarized in Table 1, from which one recognizes that the prior studies attempted to investigate the health demand for the individual. Nevertheless, unlike the previous studies, our study built a reasonable approach to measuring the unmet medical demand at the regional level, which could evaluate the allocation efficiency of the regional medical system.

Table 1. Studies for unmet medical demand.

Study

Concept

Method

Explaining variables

Grossman (1972)

Demand for health

Conceptual Framework

Medical care, age, time inputs, goods input, human capital

Blundell and Wind-meijer (2000)

Demand for health services at the ward level in the UK

Semi-parametric selection model

Average waiting time routine surgery in days, standardized estimated costs 1991–1992 acute care, NHS[①] hospital accessibility, general practitioner accessibility, the proportion of the 75 years and older not in nursing and residential homes, private hospital accessibility, standardized mortality ratio ages 0–74, standardized illness ratio ages 0–74, for residents in households only, the proportion of persons with head in manual class, proportion of those of pensionable age living alone, the proportion of dependants in single carer households, proportion of the economically active that is unemployed, proportion of residents in households with no car

Mocan et al. (2004)

Demand for medical care covered urban households.

Two-part model and a discrete factor model

Price of medical care, price of food, the opportunity cost of time, age, environmental, the variables that influence the productivity of health investment

Ariizumi (2008)

Medical care demand

Conceptual Framework

Public long-term care insurance, age, health status, medical investment, level of consumption, presence of a chronic illness

Zhou et al. (2011)

Demand for healthcare across individuals from rural China

The probit regression model, zero-truncated negative binomial equation

Outpatient price, inpatient price, income, gender, age, marriage

Pappa et al. (2013)

Unmet health needs across 1,000 consenting subjects

Multiple binary logistic regression analysis

Sex, age, marital status, children, education, occupation, urbanity, physician, consultations, chronic diseases

Chaupain-Guillot and Guillot (2015)

Unmet care needs across 400,000 individuals aged 16 and over

Multilevel logistic equation

Sex, age, self-perceived health, citizenship, education level, family situation, household income, housing tenure status, the existence of debts, whether the household has a car or not

Connolly and Wren (2017)

Unmet healthcare needs across 4922 households in Ireland

Multivariate logistic regression

Sex, age group, marital status, education, principal economic status, income, eligibility category, health status, chronic illness

Yoon et al. (2019)

Unmet medical needs across adults over 19 years old from Korea

Multiple logistic regression

Sex, age, education, marital status, economic activity, income, medical insurance type, private insurance, non-covered treatment, chronic disease, disability, regular exercise, pain, self-rated health status, depression

Fiorillo (2020)

Unmet needs for health care across 260,000 respondents from 14 Member States of the EU

Expanded probit models

Social capital, social support, individual characteristics (gender, marital status, age, household size, country of birth, education, economic features, health status, size of municipality)

Njagi et al. (2020)

Unmet need for healthcare services across 33,675 households n from Kenya

Multilevel regression model

Gender, age, education level, employment status, type of service, self-rated health, chronic illness, insurance status, household size, residence, wealth index

Jang et al. (2021)

Unmet medical needs across 2281 older adults with limited IADL from Koreans

Logistic regression analysis

Age, gender, educational level, household income, number of chronic diseases, living arrangement, contact with friend and neighbor, social activity, emotional support, instrumental support, physical support, financial support

Jung and Ha (2021)

Unmet healthcare needs across 26,598 participants aged 19 years and older from Korea

Multiple logistic regression models

Age, marital status, family member, education level, region, employment, income, occupation, medical insurance type, private insurance, smoking history, alcohol consumption, body mass index, exercise, self-rated health status, stress level, pain, depression

"

[①] NHS denote National Health Service of the United Kingdom.

Reviewer 2 Report

Thank you for the detailed replies.  No further comments.

Author Response

Many thanks for your kind help and Best Wishes.